# Fast Template Evaluation with Vector Quantization

**Mohammad Amin Sadeghi**
Department of Computer Science
University of Illinois at Urbana-Champaign
msadegh2@illinois.edu

**David Forsyth**
Department of Computer Science
University of Illinois at Urbana-Champaign
daf@illinois.edu

## Abstract

Applying linear templates is an integral part of many object detection systems and accounts for a significant portion of computation time. We describe a method that achieves a substantial end-to-end speedup over the best current methods, without loss of accuracy. Our method is a combination of approximating scores by vector quantizing feature windows and a number of speedup techniques including cascade. Our procedure allows speed and accuracy to be traded off in two ways: by choosing the number of Vector Quantization levels, and by choosing to rescore windows or not. Our method can be directly plugged into any recognition system that relies on linear templates. We demonstrate our method to speed up the original Exemplar SVM detector [1] by an order of magnitude and Deformable Part models [2] by two orders of magnitude with no loss of accuracy.

## 1   Introduction

One core operation in computer vision involves evaluating a bank of templates at a set of sample locations in an image. These sample locations are usually determined by sliding a window over the image. This is by far the most computationally demanding task in current popular object detection algorithms including canonical pedestrian [3] and face detection [4] methods (modern practice uses a linear SVM); the deformable part models [2]; and exemplar SVMs [1]. The accuracy and flexibility of these algorithms has turned them into the building blocks of many modern computer vision systems that would all benefit from a fast template evaluation algorithm. There is a vast literature of models that are variants of these methods, but they mostly evaluate banks of templates at a set of sample locations in images.

Because this operation is important, there is now a range of methods to speed up this process, either by pruning locations to evaluate a template [7, 8] or by using fast convolution techniques. The method we describe in this paper is significantly faster than any previous method, at little or no loss of accuracy in comparison to the best performing reference implementations. Our method does not require retraining (it can be applied to legacy models). Our method rests on the idea that it is sufficient to compute an accurate, fixed-precision approximation to the value the original template would produce. We use Vector Quantization speedups, together with a variety of evaluation techniques and a cascade to exclude unpromising sample locations, to produce this approximation quickly.

Our implementation is available online[1] in the form of a MATLAB/C++ library. This library provides simple interfaces for evaluating templates in dense or sparse grids of locations. We used this library to implement a deformable part model algorithm that runs nearly two orders of magnitude faster than the original implementation [2]. This library is also used to obtain an order of magnitude speed-up for the exemplar SVM detectors of [1]. Our library could also be used to speed up various convolution-based techniques such as convolutional neural networks.

As we discuss in section 4, speed comparisons in the existing literature are somewhat confusing. Computation costs break into two major terms: per image terms, like computing HOG features; and per (image×category) terms, where the cost scales with the number of categories as well as the number of images. The existing literature, entirely properly, focuses on minimizing the per (image × category) terms, and as a result, various practical overhead costs are sometimes omitted. We feel that for practical systems, all costs should be accounted for, and we do so.

## 1.1 Prior Work

At heart, evaluating a deformable part model involves evaluating a bank of templates at a set of locations in a scaled feature pyramid. There are a variety of strategies to speed up evaluation.

**Cascades** speed up evaluation by using cheap tests to identify sample points that do not require further evaluation. Cascades have been very successful in face detection algorithms (eg. [5, 6]) For example, Felzenszwalb et al. [7] evaluate root models, and then evaluate the part scores iteratively only in high-chance locations. At each iteration it evaluates the corresponding template only if the current score of the object is higher than a certain threshold (trained in advance), resulting in an order of magnitude speed-up without significant loss of accuracy. Pedersoli et al. [8] follow a similar approach but estimate the score of a location using a lower resolution version of the templates.

**Transform methods** evaluate templates at all locations simultaneously by exploiting properties of the Fast Fourier Transform. These methods, pioneered by Dubout et al. [9], result in a several fold speed-up while being exact; however, there is the per image overhead of computing an FFT at the start, and a per (image × category) overhead of computing an inverse FFT at the end. Furthermore, the approach computes the scores of all locations at once, and so is not *random-access*; it cannot be efficiently combined with a cascade detection process. In contrast, our template evaluation algorithm does not require batching template evaluations. As a result, we can combine our evaluation speedups with the cascade framework of [7]. We show that using our method in a cascade framework leads to two orders of magnitude speed-up comparing to the original deformable part model implementation.

**Extreme category scaling methods** exploit locality sensitive hashing to get a system that can detect 100,000 object categories in a matter of tens of seconds [10]. This strategy appears effective — one can't tell precisely, because there is no ground truth data for that number of categories, nor are their baselines — and achieves a good speedup with very large numbers of categories. However, the method cannot speedup detection of the 20 VOC challenge objects without significant loss of accuracy. In contrast, because our method relies on evaluation speedups, it can speed up evaluation of even a single template.

**Kernel approximation methods:** Maji and Berg showed how to evaluate a histogram intersection kernel quickly [13]. Vedaldi et al. [12] propose a kernel approximation technique and use a new set of sparse features that are naturally faster to evaluate. This method provides a few folds speed-up with manageable loss of accuracy.

**Vector Quantization** offers speedups in situations where arithmetic accuracy is not crucial (eg. [12, 14, 15, 16]). Jegou et al. [15] use Vector Quantization as a technique for approximate nearest neighbour search. They represent a vector by a short code composed of a number of sub-space quantization indices. They efficiently estimate the euclidean distance between two vectors from their codes. This work has been very successful as it offers two orders of magnitude speedup with a reasonable accuracy. Kokkinos [14] describes a similar approach to speed up dot-product. This method can efficiently estimate the score of a template at a certain location by looking-up a number of tables. Vector Quantization is our core speedup technique.

**Feature quantization vs. Model quantization:** Our method is similar to [12] as we both use Vector Quantization to speed up template evaluation. However, there is a critical difference in the way we quantize space. [12] quantizes the feature space and trains a new model using a high-dimensional sparse feature representation. In contrast, our method uses legacy models (that were trained on a low-dimensional dense feature space) and quantizes the space only at the level of evaluating the scores. Our approach is simpler because it does not need to retrain a model; it also leads to higher accuracy as shown in Table 2.

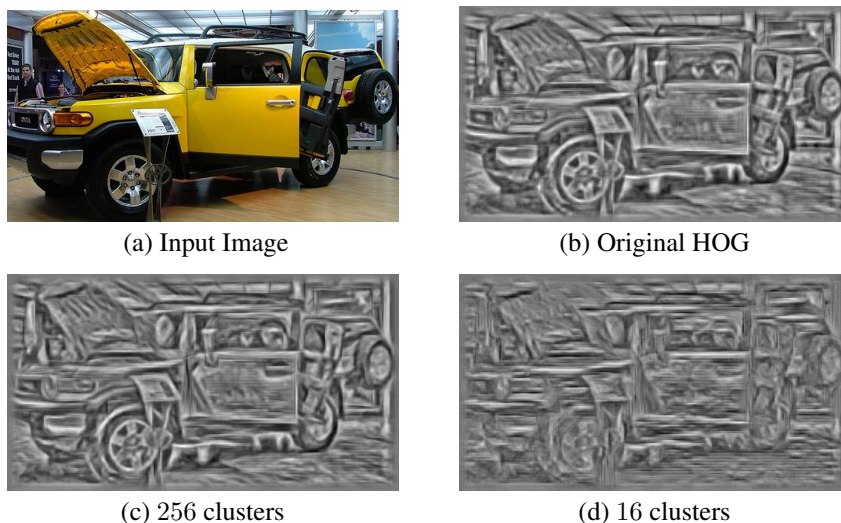

| (a) Input Image | (b) Original HOG |
| (c) 256 clusters | (d) 16 clusters |

Figure 1: Visualization of Vector Quantized HOG features. (a) is the original image, (b) is the HOG visualization, (c) is the visualization of Vector Quantized HOG feature into $c = 256$ clusters, (d) is the visualization of Vector Quantized HOG feature into $c = 16$ clusters. HOG visualizations are produced using the inverse HOG algorithm from [19]. Vector Quantized HOG features into $c = 256$ clusters can often preserve most of the visual information.

## 2 Fast Approximate Scoring with Vector Quantization

The vast majority of modern object detectors work as follows:

- In a preprocessing stage, an image pyramid and a set of underlying features for each layer of the pyramid are computed.
- For each location in each layer of the pyramid, a fixed size window of the image features spanning the location is extracted. A set of linear functions of each such window is computed. The linear functions are then assembled into a score for each category at that location.
- A post processing stage rejects scores that are either not local extrema or under threshold.

Precisely how the score is computed from linear functions varies from detector to detector. For example, exemplar SVMs directly use the score; deformable part models summarize a score from several linear functions in nearby windows; and so on. The threshold for the post-processing stage is chosen using application loss criteria. Typically, detectors are evaluated by marking true windows in test data; establishing an overlap criterion to distinguish between false and true detects; plotting precision as a function of recall; and then computing the average precision (AP; the integral of this plot). A detector that gets a good AP does so by assigning high values of the score to windows that strongly overlap the right answer. Notice that what matters here is the ranking of windows, rather than the actual value of the score; some inaccuracy in score computation might not affect the AP.

In all cases, the underlying features are the HOG features, originally described by Dalal and Triggs [3]. HOG features for a window consist of a grid of cells, where each cell contains a $d$-dimensional vector (typically $d = 32$) that corresponds to a small region of the image (typically $8 \times 8$ pixels).

The linear template is usually thought of as an $m \times n$ table of vectors. Each entry of the table corresponds to a grid element, and contains a $d$ dimensional vector $\mathbf{w}$. The score at location $(x, y)$ is given by:

$$S(x, y) = \sum_{\Delta y=1}^{m} \sum_{\Delta x=1}^{n} \mathbf{w}(\Delta x, \Delta y) \cdot \mathbf{h}(x + \Delta x - 1, y + \Delta y - 1)$$

where $\mathbf{w}$ is a weight vector and $\mathbf{h}$ is the feature vector at a certain cell (both $d$-dimensional vectors). We wish to compute an approximation to this score where (a) the accuracy of the approximation is

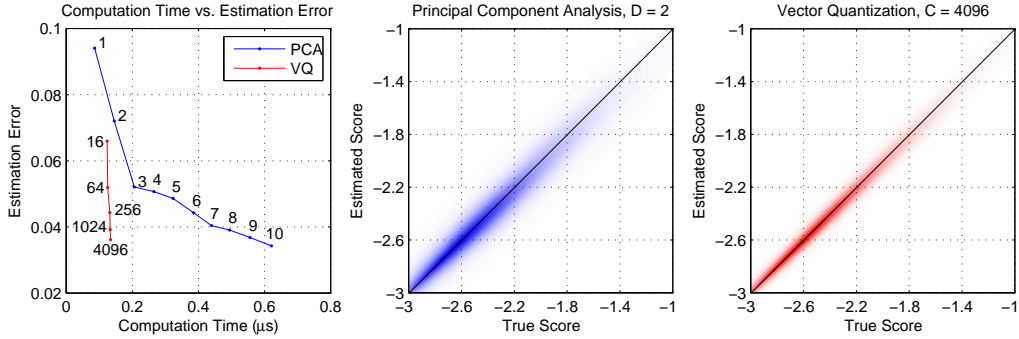

Figure 2: The plot on the left side illustrates the trade-off between computation time and estimation error $|S(x, y) - S'(x, y)|$ using two approaches: Principal Component Analysis and Vector Quantization. The time reported here is the average time required for estimating the score of a $12 \times 12$ template. The number of PCA dimensions and the number of clusters are indicated on the working points. The two scatter-plots illustrate template score estimations using $10^7$ sample points. The working points $D = 2$ for PCA and $c = 4096$ for VQ are comparable in terms of running time.

relatively easily manipulated, so we can trade-off speed and performance and (b) the approximation is extremely fast.

To do so, we quantize the feature vectors in each cell $\mathbf{h}(x, y)$ into $c$ clusters using a basic k-means procedure and encode each quantized cell $q(x, y)$ using its cluster ID (which can range from 1 to $c$). Figure 1 visualizes original and our quantized HOG features. We pre-compute the partial dot product of each template cell $\mathbf{w}(\Delta x, \Delta y)$ with all $1 \le i \le c$ possible centroids and store them in a lookup table $\mathbf{T}(\Delta x, \Delta y, i)$. We then approximate the dot product by looking up the table:

$$S'(x, y) = \sum_{\Delta y = 1}^{m} \sum_{\Delta x = 1}^{n} \mathbf{T}(\Delta x, \Delta y, q(x + \Delta x - 1, y + \Delta y - 1)).$$

This reduces per template computation complexity of exhaustive search from $\Theta(mnd)$ to $\Theta(mn)$. In practice 32 multiplications and 32 additions are replaced by one lookup and one addition. This can potentially speed up the process by a factor of 32. Table lookup is often slower than multiplication, therefore gaining the full speed-up requires certain implementation techniques that we will explain in the next section.

The cost of this approximation is that $S'(x, y) \ne S(x, y)$, and tight bounds on the difference are unavailable. However, as $c$ gets large, we expect the approximation to improve. As figure 2 demonstrates, the approximation is good in practice, and improves quickly with larger $c$. A natural alternative, offered by Felzenszwalb et al. [7] is to use PCA to compress the cell vectors. This approximation should work well if high scoring vectors lie close to a low-dimensional affine space; the approximation can be improved by taking more principal components. However, the approximation will work poorly if the cell vectors have a "blobby" distribution, which appears to be the case here. Our experimental analysis shows Vector Quantization is generally more effective than principal component analysis for speeding-up dot product estimation. Figure 2 compares the time-accuracy trade-offs posed by both techniques.

It should be obvious that this VQ approximation technique is compatible with a cascade. As results below show, this approximate estimate of $S(x, y)$ is in practice extremely fast, particularly when implemented with a cascade. The value of $c$ determines the trade-off between speed and accuracy. While the loss of accuracy is small, it can be mitigated. Most object detection algorithms evaluate for a small fraction of the scores that are higher than a certain threshold. Very low scores contribute little recall, and do not change AP significantly either (because the contribution to the integral is tiny). A further speed-accuracy tradeoff involves re-scoring the top scoring windows using the exact evaluation of $S(x, y)$. Our experimental results show that the described Vector Quantized convolution coupled with a re-estimation step would significantly speed up detection process without any loss of accuracy.

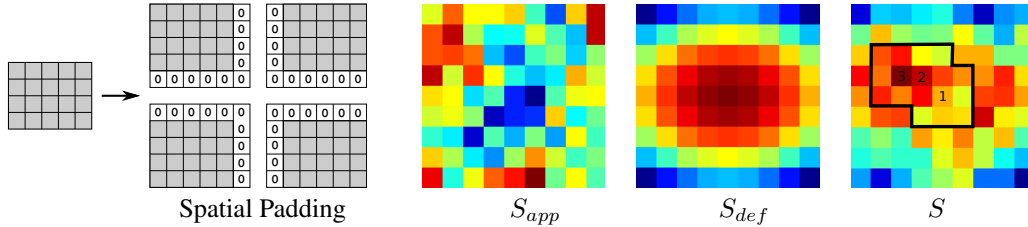

Spatial Padding $\qquad S_{app} \qquad S_{def} \qquad S$

Figure 3: Left: A single template can be zero-padded spatially to generate multiple larger templates. We pack the spatially padded templates to evaluate several locations in one pass. Right: visualization of $S_{app}$, $S_{def}$ and $S$. to estimate the maximum score we start from center and move to the highest scoring neighbour until we reach a local maximum. In this example, we take three iterations to reach global maximum. In this example we compute the template on $17$ locations in three steps (right most image).

## 3   Fast Score Estimation Techniques

Implementing a Vector Quantization score estimation is straightforward, and is the primary source of our speedup. However, a straightforward implementation cannot leverage the full speed-up potential available with Vector Quantization. In this section we describe a few important techniques we used to obtain further speed.

**Exploiting Cascades:** It should be obvious that our VQ approximation technique is compatible with a cascade. We incorporated our Vector Quantization technique into the cascade detection algorithm of [7], resulting in a few folds speed-up with no loss of accuracy. The cascade algorithm estimates the root score and the part scores iteratively (based on a pre-trained order). At each iteration it prunes out the locations lower than a certain score threshold. This process is done in two passes; the first pass uses a fast score estimation technique while the second pass uses the original template evaluation. Felzenswalb et al. [7] use PCA for the fast approximation stage. We instead use Vector Quantization to estimate the scores. In the case of deformable part models this procedure limits the process for both convolution and distance transform together. Furthermore, we use more aggressive pruning thresholds because our estimation is more accurate.

**Fast deformation estimates:** To find the best deformation for a part template, Felzenswalb et al. [7] perform an exhaustive search over a $9 \times 9$ grid of locations and find the deformation $(\Delta x, \Delta y)$ that maximizes:

$$\max_{\Delta x, \Delta y} S(\Delta x, \Delta y) = S_{app}(\Delta x, \Delta y) + S_{def}(\Delta x, \Delta y) \qquad\qquad -4 \leq \Delta x, \Delta y \leq 4$$

where $S_{app}$ is the appearance score and $S_{def}$ is the deformation score. We observed that since $S_{def}$ is convex and significantly influences the score, searching for a local minima would be a reasonable approximation. In a hill-climbing process we start from $S(0,0)$ and iteratively move to any neighbouring location that has the highest score among all neighbours. We stop when $S(\Delta x, \Delta y)$ is larger than all its $8$ neighbouring cells (Figure 3). This process considerably limits the number of locations to be processed and further speeds up the process without any loss in accuracy.

**Packed Lookup Tables:** Depending on the detailed structure of memory, a table lookup instruction could be a couple of folds slower than a multiplication instruction. When there are multiple templates to be evaluated at a certain location we pack their corresponding lookup tables and index them all in one memory access, thereby reducing the number of individual memory references. This allow using SIMD instructions to run multiple additions in one CPU instruction.

**Padding Templates:** Packing lookup tables appears unhelpful when there is only one template to evaluate. However, we can obtain multiple templates in this case by zero-padding the original template (to represent various translates of that template; Figure 3). This allows packing the lookup tables to obtain the score of multiple locations in one pass.

|  | HOG features | per image | per (image×category) | per category |
|---|---|---|---|---|
| Original DPM [2] | 40ms | 0ms | 665ms | 0ms |
| DPM Cascade [7] | 40ms | 6ms | 84ms | 3ms |
| FFLD [9] | 40ms | 7ms | 91ms | 43ms |
| Our+rescoring | 40ms | 76ms | 21ms | 6ms |
| Our-rescoring | 40ms | 76ms | 9ms | 6ms |

Table 1: Average running time of the state-of-the-art detection algorithms on PASCAL VOC 2007 dataset. The running time is braked into four major terms. Feature computation, per image pre-process, per (image×category) process and per category preprocess. The running times refer to a parallel implementation using 6 threads on a XEON E5-1650 Processor.

**Sparse lookup tables:** Depending on the design of features and the clustering approach lookup tables can be sparse in some applications. Packing $p$ dense lookup tables would require a dense $c \times p$ table. However, if the lookup tables are sparse each row of the table could be stored in a sparse data structure. Thus, when indexing the table with a certain index, we just need to update the scores of a small fraction of templates. This would both limit the memory complexity and the time complexity for evaluating the templates.

**Fixed point arithmetic:** The most popular data type for linear classification systems is 32-bit single precision floating point. In this architecture 24 bits are specified for mantissa and sign. Since the template evaluation process in this paper does not involve multiplication, the power datum would stay in about the same range so one could keep the data in fixed-point format as it requires simpler addition arithmetic. Our experiments have shown that using 16-bit fixed point precision speeds up evaluation without sacrificing the accuracy.

## 4 Computation Cost Model

In order to assess detection speed we need to understand the underlying computation cost. The current literature is confusing because there is no established speed evaluation measure. Dean et al. [10] report a running time for all 20 PASCAL VOC categories that include all the preprocessing. Dubout et al. [9] only report convolution time and distance transform time. Felzenszwalb et al. [7] compare single-core running time while others report multi-core running times.

Computation costs break into two major terms: per image terms, where the cost scales with the number of images and per (image×category) terms, where the cost scales with the number of categories as well as the number of images. The total time taken is the sum of four costs:

- **Computing HOG features** is a mandatory, per image step, shared by all HOG-based detection algorithms.

- **per image preprocessing** is any process on image data-structure except HOG feature extraction. Examples include applying an FFT, or vector quantizing the HOG features.

- **per category preprocessing** establishes the required detector data-structure. This is not usually a significant bottle-neck as there are often more images than categories.

- **per (image×category) processes** include convolution, distance transform and any post-process that depends both on the image and the category.

Table 1 compares the performance of our approach with four major state-of-the-art algorithms. The algorithms described are evaluated on various scales of the image with various root templates. We compared algorithms based on parallel implementation. Reference codes published by the authors (except [7]) were all implemented to use multiple cores. We parallelized [7] and the HOG feature extraction function for fair comparison. We evaluate all running times on a XEON E5-1650 Processor (6 Cores, 12MB Cache, 3.20 GHz).

| Method | mAP | time | Method | mAP | time |
|---|---|---|---|---|---|
| HSC [20] | 0.343 | 180s* | Vedaldi [12] | 0.277 | 7s* |
| WTA [10] | 0.240 | 26s* | DPM V4 -parts | 0.214 | 2.8s |
| DPM V5 [22] | 0.330 | 13.3s | FFLD [9] | 0.323 | 1.8s |
| DPM V4 [21] | 0.301 | 13.2s | DPM Cascade [7] | 0.331 | 1.7s |
| DPM V3 [2] | 0.268 | 11.6s | Our+rescoring | 0.331 | 0.53s |
| Rigid templates [23] | 0.31 | 10s* | Our-rescoring | 0.298 | 0.29s |

Table 2: Comparison of various different object detection methods on PASCAL VOC 2007 dataset. The reported time here is the time to complete the detection of 20 categories starting from raw image. The reference implementations of the marked (*) algorithms were not accessible so we used published time statistics. These four works were published after 2012 and their baseline computers are comparable to ours in terms of speed.

# 5   Experimental Results

We tested our template evaluation library for two well known detections methods. (a) Deformable part models and (b) exemplar SVM detectors. We used PASCAL VOC 2007 dataset that is a established benchmark for object detection algorithms. We also used legacy models from [1, 22] trained on this dataset. We use the state-of-the-art baselines published in [1, 22].

We compare our algorithm using the 20 standard VOC objects. We report our average precision on all categories and compare them to the baselines. We also report mean average precision (mAP) and running time by averaging over categories (Table 3).

We run all of our experiments with $c = 256$ clusters. We perform an exhaustive search to find the nearest cluster for all HOG pyramid cells that takes on average $76ms$ for one image. The computation of our exhaustive nearest neighbour search linearly depends on the number of clusters. In our experiments $c = 256$ is shown to be enough for preserving detection accuracy. However, for more general applications one might need to consider a different $c$.

## 5.1   Deformable Part Models

Deformable part models algorithm is the standard object detection baseline. Although there is significant difference between the latest version [22] and the earlier versions [2] various authors still compare to the old versions. Table 2 compares our implementation to ten prominent methods including the original deformable part models versions 3, 4 and 5. In this paper we compare the average running time of the algorithms together with mean average precision of 20 categories. Detailed per category average precisions are published in the reference papers.

The original DPM package comes with a number of implementations for convolution (that is the dominant process). We compare to the fastest version that uses both CPU SIMD instructions and multi-threading. All baseline algorithms are also multi-threaded. We present two versions of our cascade method. The first version (FTVQ+rescoring) selects a pool of candidate locations by quickly estimating scores. It then evaluates the original templates on the candidates to fine tune the scores. The second version (FTVQ-rescoring) purely relies on Vector Quantization to estimate scores and does not rescore templates. The second algorithm runs twice as fast with about 3% drop in mean average precision.

## 5.2   Exemplar Detectors

Exemplar SVMs are important benchmarks as they deal with a large set of independent templates that must be evaluated throughout the images. We first estimate template scores using our Vector Quantization based library. For the convolution we get roughly 25 fold speedup comparing to the baseline implementation. Both our library and the baseline convolution make use of SIMD operations and multi-threading. We re-estimate the score of the top 1% of locations for each category and we are virtually able to reproduce the original average precisions (Table 3). Including MATLAB implementation overhead, our version of exemplar SVM is roughly 8-fold faster than the baseline without any loss in accuracy.

| Method | aero | bicycle | bird | boat | bottle | bus | car | cat | chair | cow | dining table | dog | horse | motor bike | person | potted plant | sheep | sofa | train | tv | mAP | time |
|---|---|---|---|---|---|---|---|---|---|---|---|---|---|---|---|---|---|---|---|---|---|---|
| DPM V5 [22] | .33 | .59 | .10 | .18 | .25 | .51 | .53 | .19 | .21 | .24 | .28 | .12 | .57 | .48 | .43 | .14 | .22 | .36 | .47 | .39 | 0.330 | 665ms |
| Ours+rescoring | .33 | .59 | .10 | .16 | .27 | .51 | .54 | .22 | .20 | .24 | .27 | .13 | .57 | .49 | .43 | .14 | .21 | .36 | .45 | .42 | 0.331 | 21ms |
| Ours-rescoring | .26 | .58 | .10 | .11 | .22 | .45 | .53 | .20 | .17 | .19 | .21 | .11 | .53 | .44 | .41 | .11 | .19 | .32 | .43 | .41 | 0.298 | 9ms |
| Exemplar [1] | .19 | .47 | .03 | .11 | .09 | .39 | .40 | .02 | .06 | .15 | .07 | .02 | .44 | .38 | .13 | .05 | .20 | .12 | .36 | .28 | 0.198 | 13.7ms |
| Ours | .18 | .47 | .03 | .11 | .09 | .39 | .40 | .02 | .06 | .15 | .07 | .02 | .44 | .38 | .13 | .05 | .20 | .12 | .36 | .28 | 0.197 | 1.7ms |

Table 3: Comparison of our method with two baselines on PASCAL VOC 2007. The top three rows refer to DPM implementation while the last two rows refer to exemplar SVMs. We test our algorithm both with and without accurate rescoring. The two bottom rows compare the performance of our exemplar SVM implementation with the baseline. For the top three rows running time refers to per (image×category) time. For the two bottom rows running time refers to per (image×exemplar) time that includes MATLAB overhead.

## 6 Discussion

In this paper we present a method to speed-up object detection by two orders of magnitude with little or no loss of accuracy. The main contribution of this paper lies in the right selection of techniques that are compatible and together lead to a major speedup in template evaluation. The implementation of this work is available online to facilitate future research. This library is of special interest in large-scale and real-time object detection tasks.

While our method is focussed on fast evaluation, it has implications for training. HOG features require $32 \times 4 = 128$ bytes to store the information in each cell (more than $60$GB for the entire PASCAL VOC 2007 training set). This is why current detector training algorithms need to reload images and recompute their feature vectors every time they are being used. Batching is not compatible with the random-access nature of most training algorithms.

In contrast, Vector Quantized HOG features into $256$ clusters would need $1$ Byte per cell. This makes storing the feature vectors of the whole PASCAL VOC 2007 training images in random access memory entirely feasible (it would require about $1$GB of memory). Doing so allows a SVM solver to access points in the training set quickly. Our application specific implementation of PEGASOS [24] solves a SVM classifier for a $12 \times 12$ template with $10^8$ training examples (uniformly distributed in the training set) in a matter of one minute. Being able to access the whole training set plus faster template evaluation could make hard negative mining either faster or unnecessary.

There are more opportunities for speedup. Notice that we pay a per image penalty computing the Vector Quantization of the HOG features, on top of the cost of computing those features. We expect that this could be sped up considerably, because we believe that estimating the Vector Quantized center to which an image patch goes should be much faster than evaluating the HOG features, then matching.

## Acknowledgement

This work was supported in part by NSF Expeditions award IIS-1029035 and in part by ONR MURI award N000141010934.

## Footnotes

[1]http://vision.cs.uiuc.edu/ftvq

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
