[Reviews · NeurIPS 2013]

Submitted by Assigned_Reviewer_6

This paper explores a way of speeding up template evaluations in object detection by using vector quantization. The essential idea is to replace each HOG cell by an index into a dictionary. Template evaluation then amounts to a series of table lookups followed by a simple addition. In conjunction with previous work on speeding up detection [5], this paper introduces a speed up of two orders of magnitude.

Quality : The paper is well written and thoroughly evaluated. Section 4, the computational cost model, is especially well done and clears up the air vis-a-vis previous speedup papers. The timing comparisons are convincing.

Clarity : The paper is clearly written and well organized.
A suggestion (this doesn't matter for the review) : Section 4 might also gain from a graph showing how the time taken for end-to-end evaluation on a single image varies as the number of categories increases. This might be an alternative way of conveying Table 1

Originality : This is the single negative point in this review. A lot of the ideas used in this paper have actually been done before. The idea of quantizing HOG cells was actually put forward in [10] (see section 5.2). Using just the idea of quantization (without all the other additional speedups that are shown here), [10] achieves a speed-up factor of 2, while the final speedup factor that this paper demonstrates is much larger, ~100. This is a huge difference. Still, in terms of novelty, this paper is more of a somewhat novel combination of not-so-novel techniques achieving a big gain, rather than one single original idea.

Significance : This paper is extremely significant for the computer vision community, because the magnitude of the speedup is huge, and can have a big impact on downstream locations. The authors promise to release the code; if they do, then the code will very likely be quickly adopted throughout the vision community.
Summary: This paper doesn't introduce any novel machine learning techniques, but it convincingly demonstrates big speedups in an important computer vision task.

Submitted by Assigned_Reviewer_10

The paper considers HOG-based image detection. The bottleneck of such detection is the scalar product operation between the pairs of HOG cells coming from the image and the template. The central idea of the paper is to replace this scalar product with a single look up, by vector quantizing the HOG-cells on the image side (and precomputing the scalar products with cluster centers on the template side).

Pros:
+ The idea is simple, is sound and is clearly explained.

+ As HOG-based detection (including DPMs) is very popular, and the authors promice to release their code as a software library, this may become a widely used and cited paper.

+ The experimental evaluation is thorough and the results are very good for the proposed method (several-fold speedup compared to already good codes for evaluating DPMs). This good performance however required considerable micro-level optimization.

Cons:
- I can see only one major problem with the paper, but it is a big one: the central idea is already present in [10]. It is true that [10] has a different emphasis, and does retrain its models, but the central idea is there. The main difference between the proposed approach and [10] seems to be that [10] vector-quantizes both the image and the template side (ADC vs SDC in terms of [13]), hence obtaining coarser approximation.


---
if accepted, please fix the author names in the ref. [13]
Summary: A well-written paper with a natural and thouroughly-evaluated idea, which is likely to be appealing for many CV researchers. This is undermined by a strong overlap with the previously published (CVPR12) work.

Submitted by Assigned_Reviewer_11

Authors have proposed a series of 'tricks' to reduce the memory storage and computational cost of applying linear templates. The presented approximation methods achieve speed-up without significant loss of accuracy. Moreover, the authors aim for methods that trade off between speed-up and accuracy by: choosing number of quantization levels, 16-bit or 32-bit arithmetic, and finally choosing to re-score windows or not. More precisely, the key idea is to replace convolution by a look-up table. To make the latter efficient, the authors use k-means as a quantization method effectively reducing dimensionality (authors also report excellence of k-means over PCA in Fig. 2). Finally, the authors introduce different computational models showing achieved speed-up of the proposed methods over the state-of-the-art detection algorithms on Pascal VOC 2007. Similar conclusion is drawn based on the experiments on exemplar SVM.

Reference [2] (see below) seems the most related reference. It is not cited in the paper an therefore a comparison is absent. A lot of the ground is already covered in that paper - which takes away from the novelty. [2] proposes an approximation if the filters in a sparse coding way. Also nearest neightbor and pca are considered in [2].

Although the presented idea is interesting and impactful as it contributes to the speeding-up the detection task, the work is too incremental. A replacement of dot-product in the convolution operator has already been introduced in [1]. Considerable speed-up has also been shown in [2] with a sparse intermediate representation and a nearest neighbor approach where the closest part filters are retrieved. Also a pca representation has been considered there as a baseline already. The presented method scales up well as the number of classes grows. Also [3] shows significant speed-up, however [3] seems to be complementary and fits into the 'cascade' section of the paper.

Pros:
+ Authors tackle important problem of speeding up the convolution step that is at heart in many detection systems, but not only.
+ The proposed methods trades off between accuracy and speed-up
+ Simplicity of the method
+ Authors have promised publishing source code.
+ Paper is generally well written.
Cons:
- Incremental nature of the paper.
- Experiments shown in Table 1, Table 2, and Table 3 are confusing. In particular, why the paper reports different total time for the same method in different tables? Authors should also be more explicit how they have estimated the running time of unaccessible algorithms.
- It is not clear which approximation 'trick' contributed the most to the speed-up (simple look-up table, packed look-up table, 16-bit arithmetic, deformation estimates, ...). Authors should make up experiments showing the importance of every 'trick'.
- Authors should be more explicit about implementation details. For instance about the programming language being used in the experiments for every method together with the 'programming tweaks'.


[1] 'Fast, Accurate Detection of 100.000 Object Classes on a Single Machine' by T. Dean et. al.
[2] 'Sparselet Models for Efficient Multiclass Object Detection' by H. Song et. al.
[3] 'Branch and Rank: Efficient, Non-linear Object Detection' by A. Lehmann et. al.

Summary: The paper is significant as it speeds up the convolution step, however the presented idea seems to be incremental.
Author Feedback

Author rebuttal: Our paper describes a method to compute detection scores approximately 10 times faster than the fastest existing method on a CPU, with little or no loss of accuracy.

There is a general concern about novelty. Speedup papers generally will use established ideas, in polished form, and arranged so as to interact well; the innovation is in the arrangement, and the strength is in the likely impact. Our speedup will be of great practical importance, as R1, R3 point out.

Our method uses a representation similar to that of [10] (R1, R3); [10] quantizes HOG features, and so do we. But there are critical differences.

[10]'s technique is fast in principle and their highest mAP on VOC2007 is 27.7%. In contrast our full model's mAP is 33.1%. When we switch off the last level of our cascade our mAP drops to 29.8%.

[10]'s method requires the model to be retrained to accommodate their sparse feature representation. In contrast, ours is a pure computational speedup; it can use legacy models. This is an important feature of our method, because legacy models can be very highly tuned. We suffer little to no loss of accuracy over the best models, whether from DPM V4 or V5. In contrast, [10] must has based their implementation on DPM V4 combined with retraining to accommodate a novel feature representation. Their loss of accuracy may be an intrinsic property of the representation adopted for speedup. Our method does not need to accept this particular pay-off.

[10] loses about 3% mAP comparing to their baseline (30.5% to 27.7%) by retraining templates in sparse representation. Our method (without rescoring) also loses about 3% mAP (from 33.0% to 29.8%) even though we don't retrain templates (we just map regular templates into look-up tables). We show that in this particular problem transferring weights from original models works at least as accurate as retraining from scratch.

Note that [10] compares to DPM V4; we compare to DPM V5, which is state of the art in accuracy. R1, R3 note we are >20 times faster than [10]. Because our speedup is a computational speedup, it applies (as we showed) to exemplar style templates.


R2:
Q: why does the paper report different total times for the same method in different tables?
A: Page 6 line 331 points out that Table 2 reports the total time required for detecting 20 PASCAL categories. Page 7 line 347-349 points out that Table 3 reports per (image x category) time that is different from Table 2.

Q: Authors should also be more explicit how they have estimated the running time of unaccessible algorithms.
A: The total running times for four out of 12 methods (clarified with * marks in Table 2) were inaccessible on our benchmarking platform because the codes were not public. We used the the best accessible information from each work to estimate its total running times: [18] reports per category and per image running times (not the total time for 20 categories). [8] visualizes the running time on 20 categories in a plot (Figure 3.c). [21] and [10] do not explicitly report running time. [21] indicates that they are 30% faster than DPM and [10] reports they are nearly twice as fast as DPM. Note that all the four works were published in 2012 or 2013. Their processors were either not reported or were slightly faster than ours.

Q: Authors should make up experiments showing the importance of every 'trick'.
A: The speedup of a certain technique depends on which other techniques are already switched on while its effectiveness is being tested in isolation. Any pair of techniques could be orthogonal (a->2x, b->2x, a&b->4x), correlated (a->2x, b->2x, a&b->3x), redundant (a->2x, b->2x, a&b->2x), incompatible (a->2x, b->2x, a&b->NaN) or pre-requisite (a->2x, b->NaN, a&b->4x). Evaluating the effectiveness of all techniques requires studying these interactions between techniques. For speedups, ablation studies are hard to do because fast code tends not to be modular. We think the key to the best speedups is the right selection of compatible techniques. In our case the compatibility between fast template evaluation methods with cascades is the most effective part of the work (page 4, lines 188-190, 205-207). Also note that other techniques such as packed lookup tables are required to regularize memory access pattern.

Q: Authors should be more explicit about implementation details. For instance about the programming language being used in the experiments for every method together with the 'programming tweaks'.
A: All cited works use C/C++ code for their bottlenecks (e.g. BLAS, FFTW, or MATLAB mex files). Most of them use MATLAB to call major functions, as we do.

Q: Why isn't 'Sparselet Models for Efficient Multiclass Object Detection' by H. Song et. al. cited?
A: We didn't think it relevant, but could cite it if required. Note that all the work we compared was implemented on CPU; this work is implemented on GPU. They are slower than us even though they use GPU. Their best mAP is 8% lower than us.